# Diallyl Disulfide Induces Apoptosis and Autophagy in Human Osteosarcoma MG-63 Cells through the PI3K/Akt/mTOR Pathway

**DOI:** 10.3390/molecules24142665

**Published:** 2019-07-23

**Authors:** Ziqi Yue, Xin Guan, Rui Chao, Cancan Huang, Dongfang Li, Panpan Yang, Shanshan Liu, Tomoka Hasegawa, Jie Guo, Minqi Li

**Affiliations:** 1Department of Bone Metabolism, School of Stomatology Shandong University, Shandong Provincial Key Laboratory of Oral Tissue Regeneration, Jinan 250012, China; 2Department of Developmental Biology of Hard Tissue, Graduate School of Dental Medicine, Hokkaido University, Sapporo 060-8586, Japan

**Keywords:** diallyl disulfide, apoptosis, autophagy, osteosarcoma, PI3K/Akt/mTOR pathway

## Abstract

Diallyl disulfide (DADs), a natural organic compound, is extracted from garlic and scallion and has anti-tumor effects against various tumors. This study investigated the anti-tumor activity of DADs in human osteosarcoma cells and the mechanisms. MG-63 cells were exposed to DADs (0, 20, 40, 60, 80, and 100 μM) for different lengths of time (24, 48, and 72 h). The CCK8 assay results showed that DADs inhibited osteosarcoma cell viability in a dose-and time-dependent manner. FITC-Annexin V/propidium iodide staining and flow cytometry demonstrated that the apoptotic ratio increased and the cell cycle was arrested at the G_2_/M phase as the DADs concentration was increased. A Western blot analysis was employed to detect the levels of caspase-3, Bax, Bcl-2, LC3-II/LC3-I, and p62 as well as suppression of the mTOR pathway. High expression of LC3-II protein revealed that DADs induced formation of autophagosome. Furthermore, DADs-induced apoptosis was weakened after adding 3-methyladenine, demonstrating that the DADs treatment resulted in autophagy-mediated death of MG-63 cells. In addition, DADs depressed p-mTOR kinase activity, and the inhibited PI3K/Akt/mTOR pathway increased DADs-induced apoptosis and autophagy. In conclusion, our results reveal that DADs induced G_2_/M arrest, apoptosis, and autophagic death of human osteosarcoma cells by inhibiting the PI3K/Akt/mTOR signaling pathway.

## 1. Introduction

Osteosarcoma (OS) is one of the most frequent bone malignancies, developing from the bone-forming mesenchymal cell lines. The rapid expansion of OS is due to the direct or indirect formation of osteoid and osseous tissues [1]. Mortality from OS in children and adolescents remains very high, and the incidence rate reaches a second peak after the age of 60 years [2]. However, early intervention and appropriate treatments, such as chemotherapy drugs, have greatly improved the survival rate of the disease [3]. Some studies have confirmed that adjuvant chemotherapy has a beneficial effect on the relapse-free survival rate of patients with OS of the extremities [4]. However, the reality is that the drug resistance of tumors is becoming more and more complex, so new anti-cancer drugs are urgently needed.

Diallyl disulfide (DADs) is a natural organic compound in garlic and scallion, that has demonstrated anti-tumor properties in a variety of tumor types. Garlic has a long history of being used as a food additive and in pharmaceutical products. However, it was not until modern times that its anti-cancer mechanisms were demonstrated in specific studies. The effects of garlic include suppressing tumor proliferation and invasion [5], inducing G_2_/M arrest [6], and enhancing reactive oxygen species production [7]. Furthermore, DADs inhibit the growth of various tumors, such as colon cancer, bladder cancer, cervical cancer [8,9,10], and OS [5] by inducing apoptosis.

Uncontrolled cell proliferation is the most prominent feature of cancer. The integrity of the cell cycle is the basis for normal cell proliferation and is mainly regulated by cyclin dependent kinases (CDKs) and CDK inhibitor proteins. An unbalanced cell cycle promotes the occurrence and development of tumors [11,12]. Successful G_2_/M transformation is the key to cell division [13]. Many plant natural compounds inhibit tumor growth by blocking the G_2_/M phase [14,15].

Most natural organic compounds play a role by inducing cell death. Several modes of cell death have been described, such as apoptosis, autophagy, and others (necrosis and mitotic catastrophe). Apoptosis refers to the orderly death of cells controlled by genes with the purpose of maintaining homeostasis. The most notable features of apoptosis are pyknosis, DNA splitting, and the formation of apoptotic bodies [16]. Apoptosis is a strictly regulated multi-channel complex process that is mainly coordinated by activation of an aspartic acid-specific cysteine protease (caspase) cascade, including two main pathways: One relies on mitochondria (independent of the receptor) and the other involves the interaction between the death receptor and its ligand [17].

Autophagy is a process in which hydrolytic enzymes in lysosomes degrade proteins and organelles, including formation of phagophores and autophagosomes and fusion with lysosomes. The main function of autophagy is to promote cellular homeostasis. However, the role of autophagy in the tumor process is complex. Several studies have suggested that apoptosis and autophagy are interrelated and affect each other [18,19]. Autophagy can have positive or negative effects on tumor growth depending on the disease environment, and the survival function of autophagy may be harmful. In recent years, many natural organic compounds have exerted their anti-cancer activities by inducing autophagy of cells, which is of great importance to the further exploration and development of chemical anti-cancer therapy [20].

A great many anti-tumor drugs induce apoptosis and autophagy by inhibiting the AKT/mTOR pathway [21,22,23]. The PI3K/Akt/mTOR pathway is a common vulnerability in OS [24]. This signaling pathway affects most major cellular functions, so it plays a huge role in regulating basic cellular behaviors, such as growth and proliferation. The PI3K/Akt/mTOR pathway is associated with a variety of diseases, including cancer, obesity, and neurodegeneration. Early studies reported that the mTOR pathway has negative regulatory effects on apoptosis and autophagy [25,26,27]. A great deal of effort is currently being made to pharmacologically target this pathway [28].

In the present study, we explored the anti-cancer effect of DADs in OS MG-63 in vitro. In addition, we expounded on the potential mechanisms of apoptosis and autophagy through the mTOR signaling pathway.

## 2. Results

### 2.1. DADs Inhibit Osteosarcoma Cell Viability and Induces Cell Cycle Arrest at the G2/M Phase

The chemical structure of DADs is shown in Figure 1A. MG-63 cells were treated with different concentrations of DADs (0, 20, 40, 60, 80, and 100 μM) for 12, 48, and 72 h. Cell viability was measured with the Cell Counting Kit-8 (CCK-8) assay. As shown in Figure 1B, viability of OS cells treated with DADs was inhibited in a dose-and-time-dependent manner compared with the control group. The 20, 60, and 100 μM treatments were selected as representative doses for the in vitro and subsequent studies. The clone formation assay showed that the DADs treatment inhibited cloning of MG-63 cells (Figure 1C,D). DADs inhibited the colony counts of OS cells in a dose-dependent manner.

Cell cycle arrest may lead to inhibited proliferation, so we determined the effect of DADs on the cell cycle by flow cytometry. The G_0_/G_1_ phase cell population decreased, while the sub G_1_ phase and the G_2_/M phase increased significantly after treatment with 0, 20, 60, or 100 μM DADs for 24 h (Figure 1E,F).

### 2.2. DADs Induce Apoptosis of Osteosarcoma Cells

DADs may inhibit the growth of OS cells through apoptosis. Therefore, we determined whether DADs induce OS cell apoptosis through Annexin V/propidium iodide (PI) double staining. As shown in Figure 2A,B, the flow cytometry results showed that OS cells caused a dose-dependent increase in early and late apoptotic cells after the DADs treatment. We investigated the expression of important signaling proteins during apoptosis by Western blot. After a certain period of treatment, the protein expression of caspase-3, cleaved-caspase 3, and Bax increased significantly while that of Bcl-2 decreased (Figure 2C,D).

### 2.3. DADs Induce Autophagy of Osteosarcoma Cells

We continued to explore whether DADs induced autophagy in OS cells. We examined the expression of autophagy-related proteins in the DADs and control groups by Western blot analysis. The results showed that DADs increased the levels of LC3B-II, an indicator of autophagosome formation, in MG-63 cells in a dose-dependent manner relative to the controls (Figure 3A,B). Moreover, we observed an increase in p62 protein expression in the DADs-treated groups compared with that in the untreated group.

We used the autophagy inhibitor 3-methyladenine (3-MA) to perform an experiment. The 3-MA inhibits autophagosome formation during the early stage by blocking class III phosphatidylinositol 3-kinases [29]. The level of LC3-II induced by DADs in association with 3-MA (2.5 mM) was clearly less than that observed with DADs alone (Figure 3E,F). The results of Annexin V and PI double staining showed that the percentage of apoptotic cells was less in the DADs + 3-MA group than in the DADs group. However, the apoptosis rate continued to increase regardless of 3-MA compared with the control group (Figure 3C,D). The Western blot results showed that apoptosis-related proteins decreased after 3-MA treatment compared with that in the DADs only treatment groups (Figure 3G,H).

### 2.4. DADs Induces Apoptosis and Autophagy by Inhibiting the PI3K/Akt/mTOR Signaling Pathway

Previous studies have confirmed that PI3K/Akt/mTOR is a signaling pathway that has a negative regulatory effect on apoptosis and autophagy. Inhibiting the mTOR pathway promotes autophagosome formation during the early stage. Therefore, we examined whether DADs stimulates autophagy by detecting activation of mTOR. As shown in Figure 4A,B, the Western blot results indicate that after treatment with DADs for 24 h, the expression of PI3K was decreased and, at the same time, the decrease in the phosphorylation of AKT protein was observed. We further observed that the exposure of MG-63 cells to DADs decreased the phosphorylated (activated) form of mTOR as well as its downstream effectors p70S6K and p-p70S6K proteins compared with that in the control group. The CCK8 assay results showed that rapamycin (mTOR inhibitor; 100 nM) significantly increased the inhibitory effect of DADs on cell viability (Figure 4C). A Western blot analysis revealed that the level of LC3-II increased compared to that in cells treated with DADs alone, whereas there was almost no difference in the level of the p62 protein. Apoptosis-related proteins increased after rapamycin treatment compared with that in the DADs group (Figure 4D,E).

## 3. Discussion

Garlic is not only an important food seasoning, but also a traditional medicine. Epidemiological studies have shown that the consumption of garlic is inversely proportional to the incidence of cancer [30]. The medicinal value of garlic is dependent on its active ingredient garlicin. DADs, diallyl trisulfide, and diallyl tetrasulfide are the main components of garlicin. Among them, DADs has a wide range of anti-cancer effects [31]. Studies have shown that DADs inhibits the growth of various tumors by inducing apoptosis and cell cycle arrest [32,33]. 

OS has an aggressive malignant neoplasm with poor outcomes, whose cells proliferate intensively and represent very dynamic biological structure, create numerous mutations resulting in new tumor cell lines with different genotypes and phenotypes. In such malignancies, a highly variable sensitivity to therapeutics can be observed, and some cell lines develop resistance to the treatment (plant molecules including). Therefore, the biological effects of combining various plant molecules (phytochemicals) with proven cytotoxic effects administered with conventional therapy to target a substantially wider range of signaling pathways in cancer cells should be superior compared to single compounds in cancer treatment and may delay the development of resistance [34,35]. Therefore, further urgent research is needed for the identification of new molecules (including plant-derived) with excellent anticancer properties within combinational therapies against OS. In our study, we confirmed that DADs inhibited proliferation of OS cells in dose-and-time-dependent manners, caused G_2_/M phase arrest, induced apoptosis, and restricted autophagic flux in OS in vitro. We also studied the mTOR pathway mechanism during apoptosis and autophagic flux. 

Tumor development is initially manifested by uncontrolled cell division [36]. The entire process from the completion of one division to the end of the next division is called a cell cycle, which consists of interphase and mitotic periods. Interphase consists of the G_1_, S, and G_2_ phases. G_1_/S and G_2_/M transformation are two very important phases in the cell cycle. Cells in these two stages experience a complex and active period of change and are particularly susceptible to environmental conditions. In our study, flow cytometry revealed that the OS cell cycle was blocked after 24 h of DADs (0, 20, 60, 100 μM) treatment and the effect was concentration dependent. Li et al. demonstrated that the OS cell cycle could be blocked by diallyl disulfide in the G_2_/M phase [5]. This result corresponds to the CCK8 and clone forming assay results, demonstrating that DADs inhibited cell viability and proliferative activity of OS cells.

Cell cycle arrest can induce apoptosis, and the tumor inhibitory effect of DADs also includes abnormal cell death, such as apoptosis [37] and autophagy [38]. Apoptosis is composed of a complex multi-gene regulated mechanism. The caspase activation pathway plays a key role in apoptosis. HJ, K. et al. demonstrated that DADS promoted trail-mediated apoptosis by inhibiting Bcl-2 [37]. There are two pathways that have been studied most fully: The cell surface death receptor (caspase-8) pathway and the mitochondria-initiated (caspase-9) pathway. However, caspase-3 participates in both pathways [39]. Furthermore, the anti-apoptotic Bcl-2 protein prevents mitochondria from releasing cytochrome *c*, thus, keeping cells alive. Bax and Bcl-2 both belong to the Bcl-2 gene family. Bcl-2 is an inhibitor of the apoptosis gene. Bax antagonizes the inhibitory effect of Bcl-2 on apoptosis, and promotes apoptosis [40]. In our study, the caspase-3 and Bax proteins both increased, whereas the Bcl-2 protein decreased with DADs treatment. Flow cytometry showed that the proportion of apoptotic cells in the DADs treatment group increased significantly in a dose-dependent manner during the same treatment time. These results reveal that DADs stimulated caspase-dependent apoptosis.

Autophagy is a non-caspase-dependent form of cell death that degrades proteins and organelles in cells through the lysosomal pathway. Autophagy mainly plays an adaptive role in the body, protecting organisms from various pathological damage, including infection, cancer, nerve degeneration, aging and heart disease [41]. LC3 is a marker of autophagy. During autophagy, LC3-I is hydrolyzed and converted to LC3-II. Enhanced expression of autophagy leads to aggregation of P62 and complete protein degradation by fusion with a lysosome [42]. Therefore, the ratio of LC3-II/LC3-I and the p62 expression level could reflect the level of autophagy. Our experimental results show that the LC3-II/LC3-I ratio was upregulated, which confirmed that autophagy was promoted. In contrast, we observed an increase in the expression of p62 protein in our study, which is usually considered a sign of inhibited autophagy activity [29]. This observation suggested that DADs blocked the autophagic flux after early stimulation of autophagy, resulting in accumulation of the protein.

However, autophagy is a multi-stage process, and the damage and preservation of cells are unclear. We used autophagy inhibitors to further explore the mechanism of autophagy. 3-MA inhibits autophagy by inhibiting formation of the autophagosome. After adding 3-MA, LC3-II expression was downregulated and p62 expression was upregulated compared with the DADs treatment alone. Notably, the apoptosis rate of the DADs + 3-MA group decreased synchronously compared with that of the DADs alone group. We suspected that the reason is that autophagy enhances caspase-dependent cell death and independently causes cell death. However, 3-MA inhibits autophagy from the initial stage, thus reducing part of the cell death [19,43].

It is commonly known that the mTOR pathway is related to autophagy and apoptosis [26,28,44]. Our study discovered that DADs inhibits the PI3K/Akt/mTOR/p70S6K signal pathway. Inhibition of the mTOR pathway can partially activate autophagy and apoptosis. PI3K/Akt/mTOR signaling pathway is a classical pathway, which not only promotes angiogenesis and cell progress, but also plays an important role in various human malignant tumors [45]. To further test whether apoptosis and autophagy induced by DADs are related to the mTOR pathway, we added rapamycin (mTOR inhibitor). The results showed that cell proliferation in the rapamycin + DADs group was lower than that of the DADs group alone, and the levels of apoptosis-related proteins and autophagy-related proteins both increased. These finding indicate that DADs may play a role in apoptosis and autophagy through the mTOR signaling pathway. After rapamycin was used to promote autophagy, the expression of LC3-II increased, whereas content of the p62 protein did not change significantly. Thus, we conclude that DADs triggered autophagy flux; however, the process was not completed as indicated by the p62 level. This probably occurred because this process turned into apoptosis during the late stage of treatment.

However, there is no doubt that further research is needed to study the specific mechanism of DADs in OS treatment. For example, p53 is an important tumor suppressor gene, which is related to cell cycle arrest and apoptosis. The mutation and deletion of P53 gene account for about 50% of all human tumors, which leads to tumor formation, metastasis and drug resistance of tumors. MG-63 expresses P53, and p53-mediated cell death is partly related to the interaction between Bcl-2 and Bax. What’s more, the correlation between apoptosis and autophagy under DADs treatment has not been completely explained. Therefore, further exploration is needed to reveal the potential mechanism of DADs in the treatment of OS.

In conclusion, our study clarified the possible effects of DADs on OS cells and showed that DADs inhibited OS by causing G_2_/M phase arrest and inducing apoptosis and autophagy. Moreover, DADs induced apoptosis and autophagy by inhibiting the PI3K/Akt/mTOR signaling pathway. In addition, 3-MA inhibited autophagy weakened DADs-induced cell death, indicating that DADs induced autophagy-mediated cell death. The possible mechanism of action of DADs is shown in Figure 5. This study provides insight into the clinical application of this compound and a new method to treat OS.

## 4. Materials and Methods 

### 4.1. Reagents and Antibodies

Diallyl Disulfide (DADs, 30648) was purchased from Sigma-Aldrich (Shanghai, China). 3-methyladenine (3-MA, HY-19312) and rapamycin (HY-10219) were purchased from MedChemExpress (Shanghai, China). Antibodies against Bax (5023T), Bcl-2 (4223T), cleaved-caspase 3 (9664T), mTOR (2983T), p-mTOR (5536T), and p-p70S6K (108D2) were purchased from Cell Signaling Technology (CST, Shanghai, China). Anti-SQSTM1/p62 (ab109012), LC3B (ab48394), caspase-3 (ab13847), PI3K (ab180967), Akt (ab32505), p-Akt (ab192623), and p70S6K (ab32529) was purchased from Abcam (Shanghai, China). p-PI3K (AF3241) was purchased from Affinity Biosciences (Jiangsu, China). Anti-GAPDH was purchased from Proteintech (Wuhan, Hubei, China).

### 4.2. Cell Culture

The human osteosarcoma cell line MG-63 was obtained from the Cell Bank of the Chinese Academy of Sciences (Shanghai, China). Cells were raised in Dulbecco’s Modified Eagle’s Medium (DMEM/F12) supplemented with 10% fetal bovine serum (Gibco, Grand Island, NY, USA), 1% penicillin-streptomycin and 1% non-essential amino acid (NEAA) under standard culture condition (37 °C, 95% humidified air and 5% CO_2_). The medium was changed every 2 days.

### 4.3. Cell Viability Assay

For the sake of confirming the effect of DADs on OS cell proliferation, cells were incubated in 96-well plates for 24 h at a density of 5 × 10^3^ cells/well with 100 μL culture medium. Then, cells were treated with different dose of DADs (0, 20, 40, 60, 80, and 100 μM). After 24, 48, and 72 h, cell viability was detected by Cell Counting Kit-8 assay (MedChem Express, China). Adding 10 μL CCK8 working solution per well for about 2 h, the absorbance value of each well was measured at 450 nm with enzyme-labeled instrument. Then, the cell viability of the experimental group was calculated.

### 4.4. Clone Formation Assay

Cells were cultured in 6-well plates at a density of 500 cells/well and incubated under standard culture condition for 24 h. They were then treated them with different doses of DADs (0, 20, 40, 60, 80, and 100 μM), for about 9 days. Next, the medium was removed and the cell clones was washed with PBS. Shortly afterwards, they were fixed with 4 % paraformaldehyde and dyed with 0.1% crystal violet. Finally, colonies including more than 50 cells were calculated.

### 4.5. Cell Cycle Analysis

The cells were incubated in 6-well plates with a density of 2 × 10^5^ cells/well. After 24 h, cells were treated with DADs (0, 20, 60, and 100 μM) for another 24 h. Then we collected cells from 6-well plates, added 70% ethanol and fixed at 4 °C overnight. After centrifugation to remove ethanol, they were washed again with PBS and the cells were stained with propidium iodide (PI) and RNase A (KeyGEN Biotech, China). The mixture was kept at 37 °C for 15 min in the dark. Finally, the cell cycle was measured by flow cytometry, and data were analyzed with BD Accuri C6 plus (Becton Dickinson, Franklin Lakes, NJ, USA) software.

### 4.6. Apoptosis Flow-Cytometry Assay

The OS cells were seeded into 6-well plates with a density of 2 × 10^5^ cells/well, then treated with different concentrations of DADs (0, 20, 60, and 100 μM) for 24 h. Cells were collected with trypsin, washed twice with pre-chilled PBS, and then suspended with 500 μL binding buffer. Then, Annexin V-FITC and PI (Hanbio, Shanghai, China) were added respectively, and the cells were mixed. After being incubated at room temperature in the dark for 15 min, the samples were analyzed by flow cytometry using BD Accuri C6 plus (Becton Dickinson, Franklin Lakes, NJ, USA). According to the principle that phosphatidylserine can bind to Annexin V-FITC, we measured the proportion of early apoptotic cells. PI is a DNA-binding dye. It could emit red fluorescence but cannot pass through living cell membranes. The proportion of late apoptosis/dead cells can be determined by it.

### 4.7. Western Blot Analysis

Cells were scraped into EP tubes with cell scraping tools and mixed with RIPA Lysis Buffer (Betotium Institute of Biotechnology, Beijing, China). Protease and phosphatase inhibitors were added in a ratio of 100:1. All operations were carried out on ice. After measuring the protein concentration with BCA protein assays (Beyotime, Beijing, China), we added 1/4 volume of 5 × SDS loading buffer in each EP tubes and heated it at 95 °C for 5 min. Proteins were separated by 6%–15% SDS-PAGE, and transferred to the polyvinylidene fluoride (PVDF) membrane. The membranes were incubated by 1:1000~5000 primary antibodies at 4 °C overnight, followed by incubation with secondary antibodies at room temperature for 1 h. Finally, the ECL detection system (SmartChemi 420, Beijing, China) was used to measure the immune reaction zone. Each experiment was repeated at least three times.

### 4.8. Statistical Analysis

All data were represented by the average of three independent experiments. Statistical analysis was conducted with Graghpad prism 7 software (San Diego, CA, USA). Differences between experimental groups and control groups were calculated by Student’s *t-test*. *p < 0.05* was considered to have statistical significance. 

## Figures and Tables

**Figure 1 molecules-24-02665-f001:**
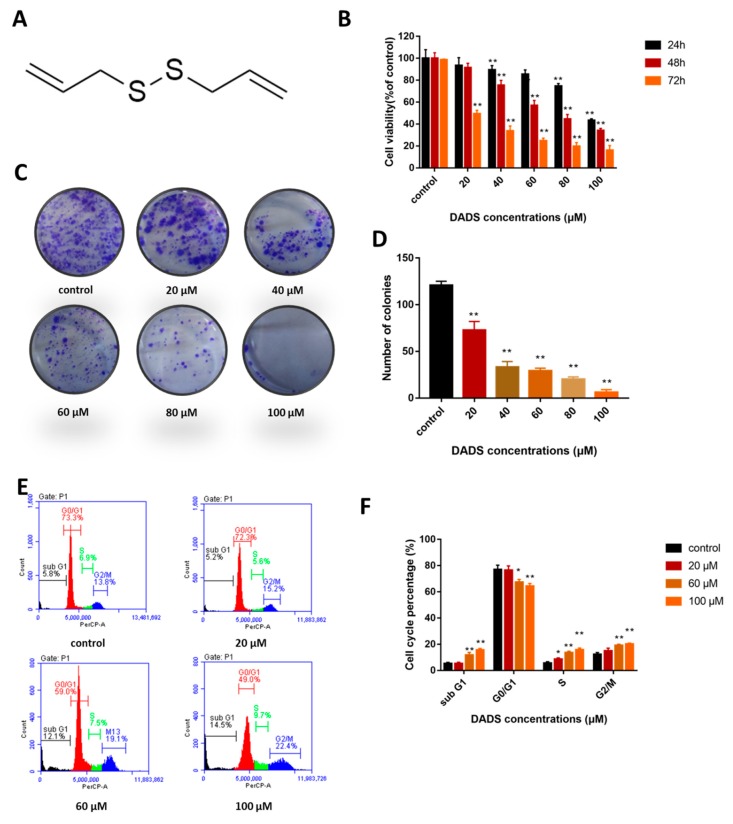
Inhibited cell proliferation and induces G_2_/M cell cycle arrest in osteosarcoma MG-63 cells. (**A**) Chemical structure of diallyl disulfide (DADs). (**B**) Cell viability. MG-63 cells were treated with the indicated dose of DADs (0, 20, 40, 60, 80, and 100 μM) for different times (24, 48, and 72 h). Cell viability was detected by CCK8 assay (*n = 3*). (**C**,**D**) Clone formation. MG-63 cells were treated with 0, 20, 60, and 100 μM DADs, and the number of cell colonies was measured by clone formation assay 9 days later. (**E**,**F**) Cells were treated with DADs for 24 h, and the cell cycle was detected by flow cytometry. G_2_/M cell cycle arrest was observed in MG-63 cells. The percentage of the sub G_1_, G_0_/G_1_, S, and G_2_/M phase cell populations were represented by the mean ± SD of at least three independent experiments. Statistical differences were analyzed by student’s *t-test* (** p* < 0.05, *** p* < 0.01 compared with control group).

**Figure 2 molecules-24-02665-f002:**
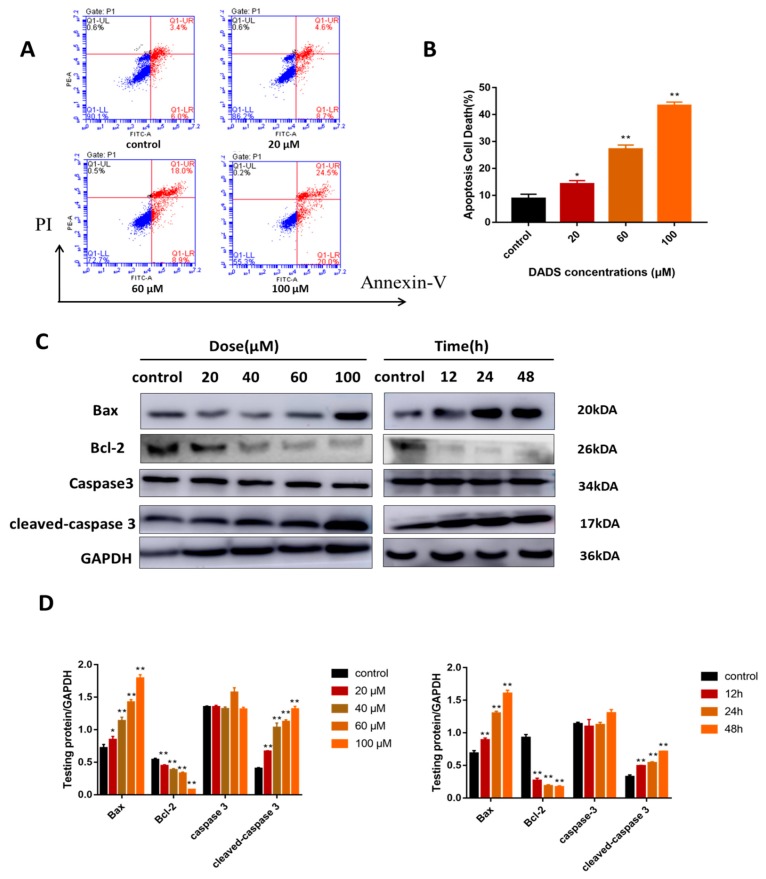
Induces caspase-dependent apoptosis in MG-63 cells. (**A**,**B**) After cells were stained by Annexin V-FITC/PI and left in dark at room temperature for 15 min, the apoptosis rate was measured by flow cytometry. Data were presented as means ± SD (*n = 3*). (**C**,**D**) Cells were treated with different doses of DADs for 24 h or incubated with DADs (60 μM) for various hours. The apoptosis-related proteins caspase-3, cleaved-caspase 3, Bax, and Bcl-2 were measured by Western blot. GAPDH was used as a loading control. (** p* < 0.05, *** p* < 0.01 compared with control group).

**Figure 3 molecules-24-02665-f003:**
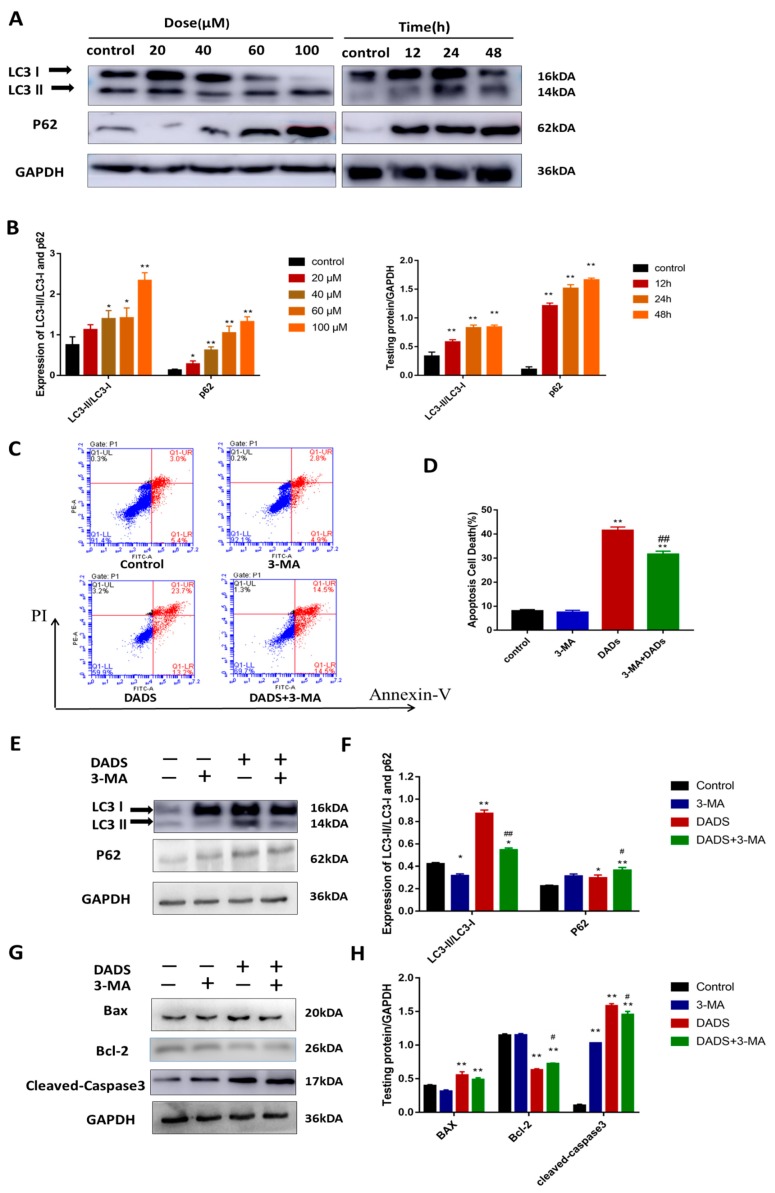
Triggered autophagy flux of MG-63 cells, and inhibition of autophagy reduces DADs-induced apoptosis. (**A**,**B**) Cells were treated with different dose of DADs for about 24 h or incubated with DADs (60 μM) for various hours. Western blotting was used to analyze protein expression, and antibodies against LC-3-I, LC3-II, and p62 were tested. (** p* < 0.05, ** *p* < 0.01 compared with control group). (**C**,**D**) Cells were pretreated with 3-MA (2.5 mM) for 2 hours and then treated with 100 μM DADs for 24 h. Apoptosis was measured by flow cytometry. The proportion of apoptotic cells from three independent experiments was shown by histograms. (**E**,**F**) Western blot showed the expression of apoptosis-related proteins LC3 and p62 with DADs or 3-MA treatment. (**G**,**H**) The Western blot results showed that caspase-3, Bax, and Bcl-2 protein expression levels after 3-MA treatment compared with that in the DADs only treatment groups. GAPDH was used as load control. (** p* < 0.05, ** *p* < 0.01 compared with the control group, *# p* < 0.05, *## p* < 0.01 compared with only DADs treated group).

**Figure 4 molecules-24-02665-f004:**
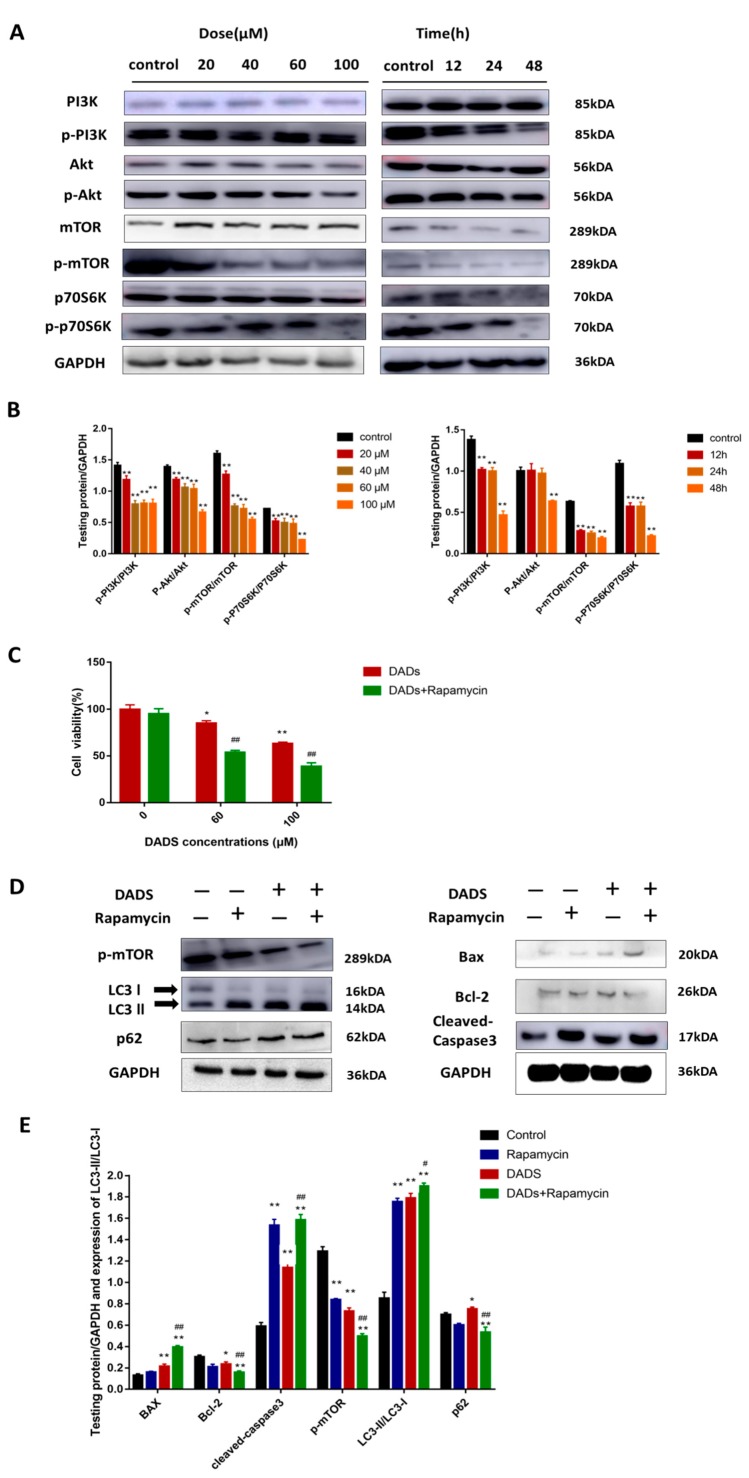
Induced apoptosis and autophagy of OS cells through mTOR pathway. (**A**,**B**) Expression of PI3K/Akt/mTOR pathway proteins were analyzed by Western blot. Cells were treated with DADs for 24 h. (**C**) Cell viability was detected by CCK8 24 h after DADs treatment (0, 60, and 100 μM). (*n = 3*). (**D**,**E**) Cells were pretreated with rapamycin (mTOR inhibitor) and then incubated with 100 μM DADs for 24 h. Western blot analyses were used to determine the levels of autophagy-related proteins (LC3-II/LC-3-I and p62), apoptosis-related proteins (caspase-3, Bax, and Bcl-2) and p-mTOR. GAPDH was used as load control. (** p* < 0.05, *** p* < 0.01 compared with the control group, *# p* < 0.05, *## p* < 0.01 compared with only DADs treated group).

**Figure 5 molecules-24-02665-f005:**
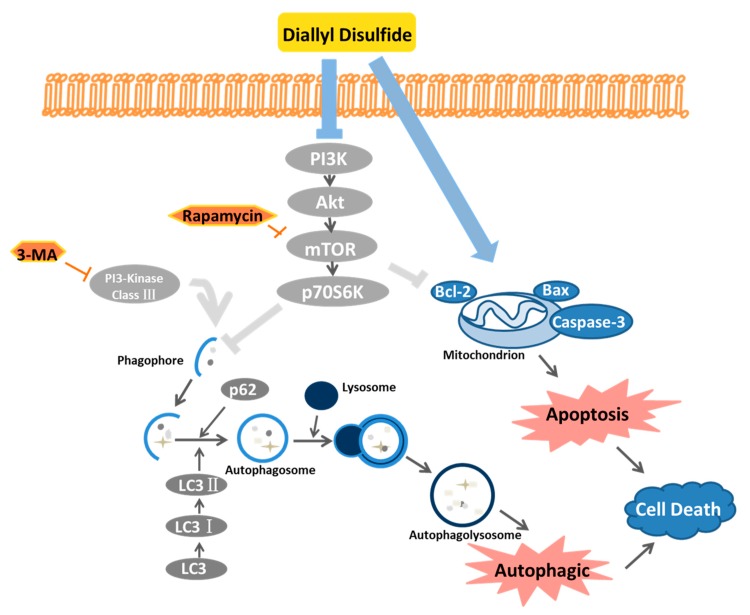
DADs inducing autophagy and apoptosis of human OS cells.

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
