# Peer review of "Diallyl Disulfide Induces Apoptosis and Autophagy in Human Osteosarcoma MG-63 Cells through the PI3K/Akt/mTOR Pathway"

_molecules, 2019, doi:10.3390/molecules24142665_

Round 1

Reviewer 1 Report

The authors Yue et al in this manuscript titled " Diallyl disulfide induces apoptosis and autophagy in human osteosarcoma MG-63 cells through the mTOR pathway" have tried to explain the efficacy of Diallyl disulfide in containing osteosarcoma cells. The author's study is based on just one cell line as the model system without any animal model such as xenograft studies. Also, the data presented is very correlative and does not have a direct mechanism that makes the presentation very un-exciting. 

The authors do not have a rational why only a single cell line is used. MG-63 express p53, but the authors have failed to address if DAD involves p53 in mediating apoptosis. Also, the use of a cell line without p53 makes this study very vague.

 The authors have not performed any mechanistic studies such as overexpression or silencing to directly point out a specific pathway being affected by DAD. Here the mTOR pathway. The authors have just barely outlined the involvement of inhibition of the mTOR pathway. The authors have not dwelled deeper into the TORC1 or TORC2 pathways.

Minor observations 

1) Figure 2C: Caspase and Cleaved caspase expression make no sense in the dose-dependent.  If the expression of cleaved caspase increases the expression of procaspase-3 must decrease. The authors need to explain this properly.

2)Figure 3E and Figure 4D require clearer LC-3 data.

3)Figure 4D: p-mTOR: the authors show that DAD does not affect the phosphorylation. But in 4A decreases phosphorylation. The authors may have to address this contradiction.  

4) Figure 4 does not show changes in pAkt and PI3K.

Reviewer 2 Report

This is original, useful, and well-done study, which use an established methods. It is evident that the article is written by experienced scientists. Manuscript is clear enough and after corrections suitable for this prestigious journal. However, several major changes must be implemented:

Abstract: Explain abbreviation 3-MA

Introduction: Paragraphs between lines 49-68 lost the linkage with the plant natural compounds – authors cannot not forget that it is the aim of this study! This linkage is very important for Introduction and deduction of the right hypothesis, please correct it seriously using suitable citations.

Aims: please mention used cancer cell line

Results are clear and well written!

Regarding Discussion, I have considerable reservations to authors: Results are poorly confronted with data of other authors; even paragraph 178-186 is without reference! Please correct.

Moreover, in the end of Discussion, I strongly suggest to insert and develop an idea which is substantially associated with the topic of this article and is important for oncology practice/clinicians – there are several sentences as suggestions (including requisite limitations of the study):

Osteosarcoma as an aggressive malignant neoplasm with poor outcomes, whose cells proliferate intensively and represent very dynamic biological structure, create numerous mutations resulting in new tumor cell lines with different genotypes and phenotypes. In such malignancies, a highly variable sensitivity to therapeutics can be observed and some of cell lines develop resistance to the treatment (plant molecules including). Therefore, the biological effects of combining various plant molecules (phytochemicals) with proven cytotoxic effects administered with conventional therapy to target a substantially wider range of signaling pathways in cancer cells should be superior compared to single compound in cancer treatment and may delay the development of resistance. Therefore, further urgent research is needed for the identification of new molecules (including plant-derived) with excellent anticancer properties within combinational therapies against osteosarcoma. Excellent references for this concept are: Kapinova et al. Biomed Pharmacother. 2017 Dec;96:1465-1477; Abotaleb et al. Biomed Pharmacother. 2018 May;101:458-477.

Line 237, please emphasize the Conclusions of the study: i.e. ...In conclusion, our study clarified the possible....

I congratulate the authors to nice results!

Reviewer 3 Report

The manuscript entitled ”Diallyl disulfide induces apoptosis and autophagy in human osteosarcoma MG-63 cells through the mTOR pathway” by Yue et al., described that diallyl disulfide (DADS) treatment induced G2/M arrest, and cell death through apoptosis and autophagy in the MG-63 osteosarcoma cells. Inhibition of phosphorylation of mTOR by DADS treatment was also demonstrated.  

Here are some comments below:

1.      The anticancer effect of DADS has been well known. Apoptosis and/or autophagy triggered by DADS was documented in various cancer cell lines. The originality of this manuscript is poor.

2.      In the Fig. 1, authors concluded that DADS induced G2/M arrest. However, based on the Fig.1E and F, DADS treatment significantly increased the proportions of both S, and G2/M phase, accompanying with decrease of the G1 proportion. These data cannot conclude that DADS induced cell cycle arrest in the G2/M phase.

3.      There are some data inconsistence in the manuscript. For example. in the Fig. 4A, and B, authors demonstrated DADS treatment significantly inhibited the phosphorylation of mTOR, and concluded that DADS inhibits the mTOR signaling pathway. However, In the Fig. 4D, and E, DADS treatment cannot inhibit the the phosphorylation of mTOR. Data inconsistence was also observed in the Fig. 3F, and Fig. 4E. In the Fig. 3E and F, the level LC3 II was significantly higher than that of LC3 I, and got a higher ratio of LC3 II/LC3 I than that of the control group. However, in the Fig. 4D and E, the ratio of LC3 II/LC3 I of the control group and that of the DADS group were comparable. These data inconsistence hugely affected the conclusion of this manuscript.

4.      There are some mistakes or questions should be corrected or answered:

    (1)       MG63 cells (should be MG-63 cells).

    (2)       In the line 87, page 2, “DADs inhibited the colony counts of OS cells in a dose-and time-dependent manner.”. However, authors only presented dose dependent data in the Fig.1C, and D. There were no time dependent data.

    (3)       The brand of the CCK8 kit was not mentioned.

Round 2

Reviewer 1 Report

Yue et al have complied to most of the comments posted in the previous section. They have also tried to honestly address their limitation. The manuscript may be accepted for publication.

The only caveat is the authors have been a little inconsistent in showing the Western Blot data. Please ensure the scans are addressed uniformly. 

Author Response

Thank you for your positive comments on our manuscript. Your comments have helped us improve the quality of the manuscript.

Reviewer 2 Report

Authors implemented all my suggestions. I recommend the publication of this manuscript.

Author Response

(The authors gave the same response as above.)

Reviewer 3 Report

The manuscript is largely improved. However, there are suggestions below:

1.      In Fig. 1F, DADs treatment significantly increases the cell populations of subG1 phase, S phase, and G2/M phase. However, in lines 91-92, authors just mentioned “……the G2/M phase increased significantly…..” It would be better to put SubG1 Phase and G2/M in the sentence.

2.      Some antibodies used in this manuscript, including PI3K, p-PI3K, AKT, p-AKT, 70S6K, and p-70S6K, were not mentioned in the section “Materials and Methods”. It should be corrected.
